# ATGen: Adversarial Reinforcement Learning for Test Case Generation

**Qingyao Li**[1]     **Xinyi Dai**[2]     **Weiwen Liu**[1*]     **Xiangyang Li**[2]
**Yasheng Wang**[2]   **Ruiming Tang**[2]   **Yong Yu**[1]     **Weinan Zhang**[1*]
[1]Shanghai Jiao Tong University   [2]Huawei Noah's Ark Lab
`{ly890306,wwliu,wnzhang}@sjtu.edu.cn`

## Abstract

Large Language Models (LLMs) excel at code generation, yet their outputs often contain subtle bugs, for which effective test cases are a critical bottleneck. Existing test generation methods, whether based on prompting or supervised fine-tuning, rely on static datasets. This imposes a "fixed-difficulty ceiling", fundamentally limiting their ability to uncover novel or more complex bugs beyond their training scope. To overcome this, we introduce ATGen, a framework that trains a test case generator via adversarial reinforcement learning. ATGen pits a test generator against an adversarial code generator that continuously crafts harder bugs to evade the current policy. This dynamic loop creates a curriculum of increasing difficulty that continuously challenges the current policy. The test generator is optimized via Reinforcement Learning (RL) to jointly maximize "Output Accuracy" and "Attack Success", enabling it to learn a progressively stronger policy that breaks the fixed-difficulty ceiling of static training. Extensive experiments demonstrate that ATGen significantly outperforms state-of-the-art baselines. We further validate its practical utility, showing it serves as both a more effective filter for Best-of-N inference and a higher-quality reward source for training code generation models. Our work establishes a new, dynamic paradigm for improving the reliability of LLM-generated code[1].

## 1 Introduction

Large Language Models (LLMs) have demonstrated remarkable capabilities in code generation (Wang & Chen, 2023; Jiang et al., 2024; Wang et al., 2025; Huang et al., 2024), tackling a wide range of programming tasks (Etsenake & Nagappan, 2024; Nijkamp et al., 2023; Li et al., 2024; 2025). However, the code they produce is often imperfect, containing subtle bugs and logical flaws (Tambon et al., 2025; Dou et al., 2024). A critical bottleneck in improving code quality through automated debugging is the scarcity of high-quality test cases that can effectively identify these errors (Dikici & Bilgin, 2025). While human-written tests are the gold standard, their manual creation is laborious and does not scale, creating a pressing need for automated test case generation (Alagarsamy et al., 2024; Chen et al., 2024a).

Initial efforts to automate test case generation based on LLMs have explored two main avenues: prompting of LLMs (Chen et al., 2022; Schäfer et al., 2023) and supervised fine-tuning (Prasad et al., 2025). One line of work involves prompting general-purpose LLMs to generate test cases based on the problem description and a given code snippet. Concurrently, more specialized approaches, exemplified by recent work like UTGen (Prasad et al., 2025), have utilized Supervised Fine-Tuning (SFT) (Shen, 2024) on pre-collected static datasets of code-test pairs. These methods aim to prompt or train a model to generate test cases with their inherient learning capability, and have shown promising initial results in this domain.

However, the unique nature of test case generation for LLM-written code presents a complex challenge that these existing approaches are ill-equipped to handle. An effective test case should satisfy two

---

[*]Corresponding authors.
[1]The resources of this work are made available at https://github.com/SIMONLQY/ATGen.

objectives (Prasad et al., 2025): 1) Output Accuracy (Ueda & Tsukada, 2021; Yang et al., 2024b), ensuring the test's (input, output) pair is correct with respect to the problem's specification. This involves a process of deriving outputs from inputs, which is inherently a complex reasoning task. Previous methods directly prompt or train on pre-collected static input-output pairs, which not only imposes a performance ceiling but also limits the model's generalization across diverse coding tasks. 2) Error-triggering (Zhong et al., 2024; Ceccato et al., 2015), or we call Attack Success, meaning ensuring the test can successfully trigger a flaw in a buggy code. This objective is inherently dynamic; its difficulty is dictated by the subtlety of the flaws in the buggy code it confronts. However, prior methods relying on static training data train the test generator on a fixed collection of buggy codes, where the types and difficulties of bugs are predetermined. Consequently, this approach imposes a fixed-difficulty ceiling on the test generator's capabilities, making the model unprepared to discover novel, more complex bugs, ultimately limiting its effectiveness as code generators become more sophisticated.

This paper provides a unique perspective on this problem. The idea is to put the test generator in an adversarial loop to train with the code generator. The code generator could provide adversarial code that challenges the current test generator's capability constantly. As the test generator improves, it forces the code generator to produce more subtle and complex bugs to evade detection. These new, harder-to-find bugs, in turn, serve as a dynamic curriculum that pushes the test generator beyond its current capabilities, effectively breaking the fixed-difficulty ceiling inherent in static methods.

In light of this, we introduce ATGEN (Adversarial Test Generator), a novel framework that trains a test generator via Reinforcement Learning (RL) within an adversarial loop. We leverage RL to move beyond static mimicry, allowing the generator to learn to reason from trial-and-error and learn a dynamic policy that explicitly optimizes the trade-off between output accuracy and attack success. To break the fixed-difficulty ceiling, we introduce an adversarial code generator into that RL process that creates a dynamic curriculum. Instead of training on a fixed set of bugs, our test generator is continuously challenged by new, "hard" buggy code that is specifically generated to evade detection by the current policy. This adversarial setup forces the test generator to constantly improve and uncover progressively more subtle flaws. Our main contributions can be summarized as follows:

- We propose a novel RL-based framework for test generation that trains a test case generator to reason and dynamically navigate the optimization of output accuracy and attack success, outperforming static prompting and SFT approaches.

- We introduce an adversarial training paradigm where the test generator is trained against a code generator, enabling it to discover more complex and subtle code flaws than with a static dataset.

- We demonstrate the practical utility of ATGEN in both the inference and training of code generation, showing that its generated tests serve as both a more effective filter for Best-of-N inference and a higher-quality reward source for RL-based code generator training.

Extensive experiments show that ATGEN significantly outperforms strong baselines. Our method demonstrates a superior ability to balance its core objectives, especially on difficult problems, establishing a new and more effective paradigm for automated test generation.

## 2 RELATED WORK

**Automated Test Case Generation with LLMs**  The advent of Large Language Models (LLMs) has shifted the focus of automated test case generation from traditional structural coverage (Huang et al.; Wu et al., 2025; Zhang et al., 2023) to addressing the unique logical flaws in AI-generated code (Huang et al.; Wu et al., 2025; Zhang et al., 2023). These modern approaches primarily fall into two categories: prompting-based methods (Chen et al., 2022; 2024b; Yang et al., 2024a) and fine-tuning-based methods (Prasad et al., 2025). Prompting-based methods (Chen et al., 2024b) leverage prompt engineering to guide general models like GPT-4 (Achiam et al., 2023), but are often limited by their unspecialized reasoning capabilities. In contrast, fine-tuning-based methods train specialized models on curated datasets. The most direct precursor, UTGen (Prasad et al., 2025), uses Supervised Fine-Tuning (SFT) to balance generating bug-revealing inputs ("attack rate") with predicting correct outputs ("output accuracy"). However, these approaches are fundamentally constrained by their reliance on static data. UTGen, for instance, is limited by its static training dataset, which prevents it from adapting to find novel or more complex bugs beyond what it has already seen.

**Reinforcement Learning for Code-related Tasks**  Reinforcement Learning (RL) has emerged as a powerful paradigm for code-related tasks, using feedback from the execution environment (e.g., unit tests) to optimize models beyond standard supervised learning. A significant line of work has applied RL to improve code generation. Approaches like CodeRL (Le et al., 2022) and others (Pan et al., 2023; Gou et al., 2023; Pan et al., 2024) established the viability of using unit test feedback as a reward signal, while Deepseek-R1 (Guo et al., 2025) demonstrated that RL can enhance an LLM's general reasoning and coding abilities. More recently, RL has been applied to automated program repair. Frameworks such as Repair-R1 (Hu et al., 2025) co-optimize test generation and bug repair, while others like Repairity (Tang et al., 2025) use feedback from an LLM judge. These works highlight the power of RL in training agents that modify code. In contrast, ATGEN aims to create a powerful, standalone test generator. Instead of learning a policy to write correct code against a fixed verifier, ATGEN learns a policy to explore the input space to falsify a given program, thereby providing a high-quality reward signal for any downstream agent.

## 3 PRELIMINARIES

In this work, we focus on the task of *automated test case generation*. Formally, given a code problem description $Q$ and a potentially faulty code implementation $C_{\text{buggy}}$, the goal is to train a test generator, represented by a policy $\pi_\theta$, to generate a unit test include input $x$ and output $y$, $T_{\text{gen}} = (x, y) = \pi_\theta(Q, C_{\text{buggy}})$. An effectively generated test case must satisfy two key objectives:

- **Output Accuracy:** The generated output $y$ must be correct with respect to the problem description $Q$. This is formally verified by checking if $y = C_{\text{gold}}(x)$, where $C_{\text{gold}}$ is a ground-truth code solution.

- **Attack Success:** The generated test case must reveal the flaw in the faulty code, meaning the execution of $C_{\text{buggy}}$ on input $x$ does not yield the correct output $y$, i.e., $C_{\text{buggy}}(x) \neq y$.

It is evident that successfully accomplishing this task demands a level of reasoning capability comparable to that required in code generation. The model must directly predict the output corresponding to a given input, which necessitates a thorough understanding of the input–output mapping—precisely the objective of code or program synthesis. Moreover, unlike standard code generation, test generation requires analyzing a potentially buggy code snippet and producing targeted test cases to expose its flaws. This places an additional requirement on the model: the ability to identify subtle code errors. In summary, the key challenges lie in enhancing the model's reasoning capacity for this task and equipping it with the capability to "attack" buggy code with subtle defects.

## 4 ATGEN

We propose ATGEN, an adversarial reinforcement learning framework that trains a robust test generator through a dynamic, self-improving curriculum. As illustrated in Figure 1, our framework is composed of two interconnected parts: an **RL-based Test Generator Training** part that optimizes the test generator, and an **Adversarial Code Generation** part that dynamically creates challenging training data.

### 4.1 RL-BASED TEST GENERATOR TRAINING

**RL Formulation.**  The core training of our framework is a test generator trained via reinforcement learning so that the model can learn to optimize output accuracy and attack success rate directly. We formalize this process as follows:

- **State ($s_t$):** The state is a tuple consisting of the problem description and the current buggy code, $s_t = (Q, C_{\text{buggy}})$.

- **Action ($a_t$):** The action is the generated test case, an I/O pair $a_t = T_{\text{gen}} = (x, y)$.

- **Policy ($\pi_\theta$):** The test generator is modeled as a stochastic policy $\pi_\theta(a_t|s_t)$, parameterized by $\theta$, which we aim to optimize.

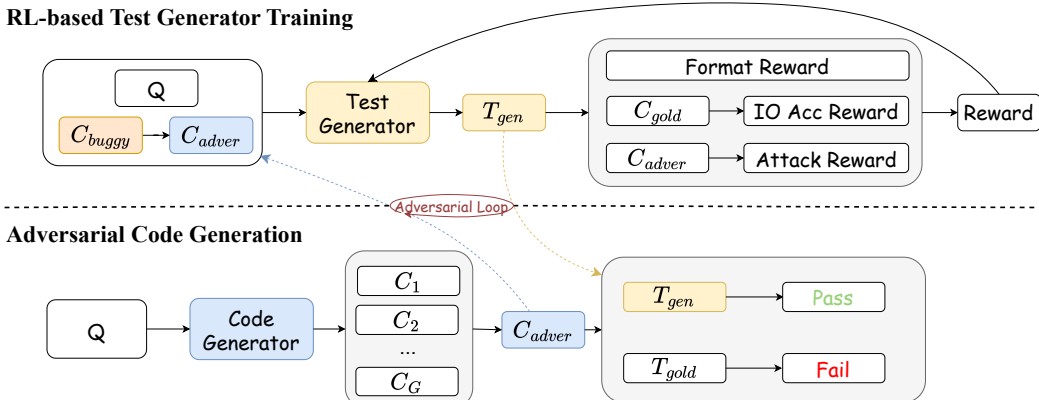

Figure 1: The overall architecture of the ATGEN framework. The top panel shows the core RL training loop: the test generator (policy) receives a state $(Q, C_{\text{adver}})$ and generates a test case $T_{\text{gen}}$ (action). It then receives a multi-component reward. The bottom panel shows the adversarial data generation loop: a code generator is tasked to sample a new, harder adversarial code $C_{\text{adver}}$ that passes the current $T_{\text{gen}}$ but fails against a ground-truth test suite $T_{\text{gold}}$. This new $C_{\text{adver}}$ is then fed back into the training loop, creating a dynamic curriculum.

At each step, the policy model (test generator) attempts to produce an action in the form of a test case (i.e., an input-output pair), based on the given question and the buggy code. The goal is for this test case to be both valid and capable of triggering faulty behavior in the buggy code, such as causing it to crash or produce incorrect output. Upon receiving the reward signal, the test generator updates its policy by computing the loss according to the chosen reinforcement learning algorithm. In this work, we employ GRPO (Shao et al., 2024), an actor-only method that eliminates the need for a separate critic model, thereby reducing computational and memory overhead.

**Test Generation Reward Function.** To effectively guide the policy, we design a multi-component reward function $R_t$ that explicitly captures the desired properties of a good test case. As shown in the top panel of Figure 1, the total reward is a weighted sum of three components:

- **IO Acc Reward** $R_{\text{acc}}$: It represents the correctness of the IO pair, which is calculated by executing a gold code $C_{\text{gold}}$ on the generated input $x$ and comparing it to the generated output $y$.
- **Attack Reward** $R_{\text{attack}}$: It is positive if the buggy code $C_{buggy}$ fails on the generated test case, either raising an execution error or output differently from the generated output. It can only be required when the IO pair is correct.
- **Format Reward** $R_{\text{format}}$: To activate the model's thinking ability, referencing Deepseek-R1's (Guo et al., 2025) training format setting, the reasoning process and answer are enclosed within `<think>` and `</think>` and `<answer>` and `</answer>` tags, respectively.

The final reward is a weighted sum of multiple rewards:

$$R_t = w_{\text{acc}} \cdot R_{\text{acc}} + w_{\text{attack}} \cdot R_{\text{attack}} + w_{\text{format}} \cdot R_{\text{format}}, \tag{1}$$

where the weights $w$ are hyperparameters that balance the different objectives during training. Each reward is within the range of $[-0.5, 1.0]$ and it's set to $-0.5$ when the form is not right.

## 4.2 ADVERSARIAL CODE GENERATION

While the reinforcement learning framework allows the test generator to directly optimize for output accuracy and attack success, its true potential is constrained by the static nature of the buggy codes it trains on. Training on a fixed collection of bugs, where difficulties are predetermined, imposes the fixed-difficulty ceiling inherent in prior methods. An agent trained in such a static environment cannot learn to overcome novel, more complex challenges. To break this ceiling and unlock the full potential of RL, the training environment itself must evolve alongside the agent. To this end, ATGEN incorporates an adversarial loop to dynamically generate a curriculum of increasingly challenging buggy code, ensuring the test generator is continuously pushed beyond its current capabilities.

**The Adversarial Process.** As shown in the bottom panel of Figure 1, this part functions as a data augmentation engine that creates "hard" training instances dynamically for our test generator. For a given problem $Q$ and a test case $T_{\text{gen}}$ produced by our current policy, we prompt a separate code generator model. This generator is tasked with producing a new adversarial code, $C_{\text{adver}}$, that satisfies two critical conditions:

- It must remain **incorrect**, so that it is a buggy code for the test generator to attack, meaning it fails against the full ground-truth test suite, $\exists (x', y') \in T_{\text{gold}}$ s.t. $C_{\text{adver}}(x') \neq y'$.

- It must **pass** the generated test case $T_{\text{gen}}$, meaning $C_{\text{adver}}(x) = y$. This ensures the new bug is not detectable by the current test generator's generation.

This process generates buggy code that is specifically designed to be challenging for the current iteration of the test generator.

**Unconditional and Adaptive Modes.** There are two primary strategies for obtaining $C_{\text{adver}}$ from the code generator. A straightforward approach is to directly instruct the generator to produce code that satisfies the adversarial criteria (i.e., passing a specific generated test while remaining globally incorrect). While this method is computationally inexpensive, it risks introducing a distributional shift; the bugs in the resulting code are deliberately engineered rather than being natural bugs. Training on such synthetic artifacts could mislead the test generator into learning to detect artificial, rather than realistic flaws.

Therefore, we adopt a more robust, *sampling-based* approach. In this paradigm, we provide the code generator with only the problem description $Q$ and sample multiple code potential solutions. From this pool of naturally generated outputs, we filter for instances that coincidentally meet the adversarial criteria. This ensures that the resulting $C_{\text{adver}}$ contains authentic bugs, providing a more realistic and challenging training curriculum for the test generator.

However, the significant computational cost of sampling for every training instance motivates a more nuanced strategy. We therefore propose and evaluate two distinct modes for adversarial data generation:

- **Unconditional Mode:** We unconditionally generate a new adversarial code $C_{\text{adver}}$ via sampling to replace the original buggy code $C_{\text{buggy}}$ for every instance in the training batch, making every $C_{\text{buggy}}$ replaced by $C_{\text{adver}}$.

- **Adaptive Mode:** To improve efficiency, we conditionally trigger the sampling process. A new $C_{\text{adver}}$ is generated only if the current test generator can already successfully attack the original $C_{\text{buggy}}$. If the original bug is already challenging enough to evade detection, we reuse it, conserving computational resources for cases where they are most needed.

**Dynamic Curriculum.** These newly generated, harder pairs of $(Q, C_{\text{adver}})$ then replace the original $(Q, C_{\text{buggy}})$ to the training data pool for the test generator. This creates a self-improving ecosystem where, as the test generator becomes more adept at finding certain types of bugs, the adversarial code generator is forced to select code with more subtle and complex flaws. This dynamic curriculum ensures that the test generator is continuously challenged and learns to identify a wider and more difficult range of programming errors than would be possible with a static dataset.

## 5 EXPERIMENTS

### 5.1 EXPERIMENTAL SETUP

**Datasets.** We train and evaluate our test generator on a subset of 3000 problems from APPS (Hendrycks et al., 2021) and Codeforces (MatrixStudio, 2024). We used GPT-4o-mini (OpenAI, 2024) to sample buggy codes for each problem, creating a training set of 16,822 and a test set of 911 (Question, buggy code) pairs. To analyze performance on bugs of varying subtlety, we partitioned this benchmark into Easy, Medium, and Hard tiers based on an initial evaluation of bug-finding difficulty. We use Qwen2.5-7B-Instruct (Qwen et al., 2024) to initially attack them by generating test cases. Then we rank them by attack success rate, and then divide the ranked set into three equal parts. For

each problem, we keep a ground-truth solution ($C_{\text{gold}}$) to verify the correctness of generated test case outputs and a suite of human-written tests ($T_{\text{gold}}$) to validate the incorrectness of adversarial code.

**Baselines.** We compare ATGEN against two categories of methods:

- **Prompting-based Methods**: We evaluate large language models that have demonstrated strong general reasoning and coding abilities. This includes GPT-4o (Hurst et al., 2024), GPT-4o-mini (OpenAI, 2024), GPT-4-turbo (Achiam et al., 2023), instruction-tuned versions of the Qwen2.5 series (Qwen et al., 2024) and Qwen3 series (Yang et al., 2025). These models are prompted to generate test cases without any specific fine-tuning for the task.
- **Supervised Fine-Tuning (SFT) Method**: We include UTGen (Prasad et al., 2025), the state-of-the-art SFT-based approach for unit test generation. We evaluate both the 3B and 7B parameter versions of UTGen to provide a comprehensive comparison against prior specialized methods.

**Evaluation Metrics.** We assess performance using two primary metrics:

- **IO Accuracy (%):** The percentage of generated test cases $(x, y)$ that are correct, i.e., $y = C_{\text{gold}}(x)$.
- **Attack Rate (%):** The percentage of test cases that are first confirmed correct (passes the IO Accuracy check) and then causes a buggy code to fail, either by crashing or producing an incorrect output (i.e., $C_{\text{buggy}}(x) \neq y$).

**Implementation Details.** Our ATGEN framework is built on Qwen2.5-3B-Instruct and Qwen2.5-7B-Instruct (Qwen et al., 2024) backbones, using the veRL (Sheng et al., 2025) framework with GRPO as the RL algorithm. The adversarial code generation loop uses GPT-4o-mini (OpenAI, 2024) as a separate code generator to produce challenging buggy code samples. All reward components $w_{\text{acc}}, w_{\text{attack}}, w_{\text{format}}$ are weighted equally. And if not specified, we use adaptive mode for ATGEN for our analysis experiments. The data used for SFT baselines and RL methods are exactly the same split.

## 5.2 MAIN RESULTS: ADVERSARIAL TRAINING FOR TEST CASE GENERATION

We present the main results in Table 1. Our findings consistently demonstrate the superiority of our proposed methods:

**ATGEN Framework Establishes a New State-of-the-Art.** Our ATGEN establishes a new state-of-the-art, significantly outperforming existing methods in both IO Acc and Attack Rate. Our best-performing model, using the Qwen2.5-7B-Instruct backbone, achieves a nearly 60% relative improvement in Attack Rate over the strongest proprietary baseline, GPT-4-turbo, and is more than twice as effective at finding bugs as the prior method, UTGen (7B) (36.99% vs. 16.24%). These results demonstrate the high effectiveness of combining RL with a dynamic adversarial curriculum.

**Reinforcement Learning Provides a Major Performance Leap.** To isolate the benefits, we also evaluated a non-adversarial version of our framework. Even this simplified model on its own constitutes a major advance over prior work, which may be due to the stimulation of the model's reasoning ability by RL training (Yue et al., 2025; Xie et al., 2025). This substantial performance leap validates that using RL to directly optimize for bug detection and output correctness is a far more effective paradigm than static fine-tuning.

**Different Modes Achieve Superiority Depending On Backbone Models.** Our Unconditional and Adaptive modes show that the optimal strategy is determined by the scale of the backbone model. For the larger 7B model, the targeted curriculum of the Adaptive mode is more effective at maximizing the Attack Rate. Conversely, for the smaller 3B model, the Unconditional mode's constant stream of diverse challenges yields a higher Attack Rate. This suggests that while more capable models benefit from focused challenges, less capable ones may learn varied attack patterns more effectively from a continuous curriculum.

## 5.3 ANALYSIS OF THE ACCURACY-ATTACK TRADE-OFF

To understand the accuracy-attack trade-off, we introduce *Input Attack Rate* metric. It is calculated by taking the generated input, pairing it with the ground-truth output from a gold

| | **Total** | | **Easy** | | **Medium** | | **Hard** | |
|---|---|---|---|---|---|---|---|---|
| **Method** | IO Acc (%) | Attack Rate (%) | IO Acc (%) | Attack Rate (%) | IO Acc (%) | Attack Rate (%) | IO Acc (%) | Attack Rate (%) |
| **Baselines** | | | | | | | | |
| GPT-4o | 34.02 | 20.63 | 24.42 | 19.47 | 35.85 | 23.02 | 41.77 | 19.40 |
| GPT-4-turbo | 41.16 | 23.38 | 34.32 | 26.73 | 42.10 | 23.35 | 47.03 | 20.06 |
| GPT-4o-mini | 34.57 | 17.23 | 27.72 | 21.12 | 35.19 | 17.76 | 40.78 | 12.82 |
| Qwen2.5-3B-Instruct | 14.05 | 6.58 | 15.51 | 12.21 | 16.12 | 5.26 | 10.52 | 2.30 |
| Qwen2.5-7B-Instruct | 26.56 | 14.37 | 23.43 | 19.80 | 28.28 | 15.78 | 27.96 | 7.56 |
| Qwen2.5-32B-Instruct | 35.01 | 21.62 | 29.70 | 24.09 | 36.51 | 24.01 | 38.81 | 16.77 |
| Qwen3-4B | 28.64 | 16.79 | 27.39 | 22.44 | 32.23 | 17.76 | 26.31 | 10.91 |
| Qwen3-8B | 38.19 | 22.72 | 38.28 | 33.66 | 39.80 | 23.35 | 36.51 | 11.18 |
| Qwen3-32B | 37.87 | 17.89 | 32.01 | 25.08 | 41.18 | 18.09 | 40.46 | 10.52 |
| UTGen (3B) | 22.83 | 12.29 | 23.10 | 18.81 | 25.00 | 12.92 | 20.39 | 5.26 |
| UTGen (7B) | 31.83 | 16.24 | 28.71 | 22.77 | 32.56 | 17.43 | 34.21 | 8.55 |
| **Ours (Backbone: Qwen2.5-3B-Instruct)** | | | | | | | | |
| ATGEN (w/o Adver) | 66.95 | 29.96 | 67.98 | 40.92 | 68.09 | 29.93 | 64.80 | 19.07 |
| ATGEN (Unconditional) | 68.93 | 32.38 | 68.64 | 42.90 | 73.35 | 32.89 | 64.80 | **21.38** |
| ATGEN (Adaptive) | 70.36 | 30.95 | 68.31 | 40.92 | 73.68 | 32.56 | 69.07 | 19.40 |
| **Ours (Backbone: Qwen2.5-7B-Instruct)** | | | | | | | | |
| ATGEN (w/o Adver) | 71.56 | 34.02 | 71.62 | 47.85 | 73.63 | 35.85 | 69.40 | 18.42 |
| ATGEN (Unconditional) | **74.97** | 34.57 | 70.95 | 47.19 | **79.27** | 36.84 | **74.67** | 19.73 |
| ATGEN (Adaptive) | 74.42 | **36.99** | **76.23** | **51.15** | **79.60** | **38.81** | 67.43 | 21.05 |

Table 1: Intrinsic evaluation of test case generation methods on a subset of APPS and Codeforces benchmarks. Our RL-trained models, ATGEN, significantly outperform SFT-based (UTGen) and prompting baselines. Best results in each column for our methods are in **bold**.

code, and then checking if this corrected IO pair successfully fails the buggy code. It measures a generated input's raw bug-finding capability, which reveals a fundamental trade-off: test inputs effective at finding bugs (high *Input Attack Rate*) are often corner cases, making it difficult for the model to predict their correct output, thus leading to lower *IO Accuracy*.

| Reward Configuration | IO Acc (%) | Attack Rate (%) | Input Attack Rate (%) |
|---|---|---|---|
| **IO Acc + Input Attack** | 44.67 | 30.07 | **62.56** |
| **Attack Rate Only** | **67.72** | 29.74 | 47.53 |
| **Three Combined** | 65.64 | **30.29** | 47.09 |

Table 2: Analysis of different reward configurations for our non-adversarial RL model, ATGEN (w/o Adver). Results highlight the trade-off between IO Accuracy and the raw attacking potential of generated inputs (Input Attack Rate).

**Trade-off by Different Reward Settings.** We first investigate whether this trade-off can be navigated by simply engineering the reward function. We trained variants of ATGEN (w/o Adver) with different reward compositions: one that rewards only the final, correct and valid attack (*Attack Rate Only*); one that jointly rewards output correctness and raw input attack capability (*IO Acc + Input Attack*); and a balanced version with all reward components (*Three Combined*).

The results, presented in Table 2, reveal a clear and informative trade-off. When explicitly rewarding for *Input Attack Rate*, the model achieves the highest score on that metric but at a significant cost to its *IO Accuracy*. Conversely, the configuration rewarding only the final *Attack Rate* yields the highest *IO Accuracy*. This suggests the model learns that a prerequisite for achieving the final attack reward is to first generate a correct I/O pair, thus adopting a more conservative but accurate strategy. Most importantly, across all configurations, the final usable *Attack Rate* remains largely stagnant at around 30%. This indicates that while reward engineering can shift the model's focus, it does not fundamentally enhance its ability to generate tests that are attacking and correct. This limitation motivates the need for a more advanced, adversarial training paradigm, which we explore next.

**Better Trade-off by ATGEN.** To rigorously evaluate the impact of adversarial training, we analyze the performance of ATGEN and its non-adversarial counterpart across various training configurations. We pick key configurations of GRPO, such as the number of samples per optimization step (e.g., 128 vs. 64) and the GRPO group generation number (e.g., 6 vs. 8). A smaller sample count per optimization step corresponds to a more "online" learning setting. The results, presented in Table 3.

For the non-adversarial model ATGEN (w/o Adver), it is forced into a suboptimal trade-off under different training dynamics, often sacrificing IO Accuracy for a higher Input Attack Rate, illustrating

that without a dynamic curriculum, pushing the model to find more challenging inputs can severely degrade its ability to predict correct outputs.

Table 3: Comparison of IO Accuracy and Input Attack Rate across different hyperparameter configurations. The proposed ATGEN framework demonstrates a clear performance improvement over its non-adversarial counterpart.

| Hyperparameter Set | Metric | ATGEN (w/o Adver) | ATGEN | Δ |
|---|---|---|---|---|
| (128, 6) | IO Accuracy (%) | 71.56 | **74.09** | **+2.53** |
| | Input Attack Rate (%) | 46.87 | **47.99** | **+1.12** |
| (64, 6) | IO Accuracy (%) | 73.76 | **74.96** | **+1.20** |
| | Input Attack Rate (%) | 47.63 | **49.17** | **+1.54** |
| (64, 8) | IO Accuracy (%) | 69.59 | **75.30** | **+5.71** |
| | Input Attack Rate (%) | 49.06 | **47.85** | **-1.21** |

In contrast, the full ATGEN framework consistently establishes a superior performance frontier. In the (128, 6) and (64, 6) configurations, ATGEN simultaneously improves both metrics, delivering a clear win-win. Most tellingly, in the (64, 8) setting where the baseline falters, ATGEN demonstrates its robust balancing capability. It achieves a massive +5.71% absolute gain in IO Accuracy while keeping the Input Attack Rate highly competitive. This shows that the dynamic adversarial loop prevents the model from overfitting to a single metric, enabling it to learn a more generalizable policy.

## 5.4 DOWNSTREAM APPLICATION: ENHANCING CODE GENERATION

To assess the practical utility of our trained test generators, we evaluate their effectiveness in both the inference and training of code generation.

**ATGEN as a Best-of-N Filter.** We evaluate our test generator's utility for inference in a Best-of-N setting on the APPS dataset. For each problem, we sample N candidate solutions and use the test generator to create a suite of $k_{test}$ unit tests. The candidate code with the highest pass rate against this suite is selected, and its final performance is measured on a private ground-truth test set.

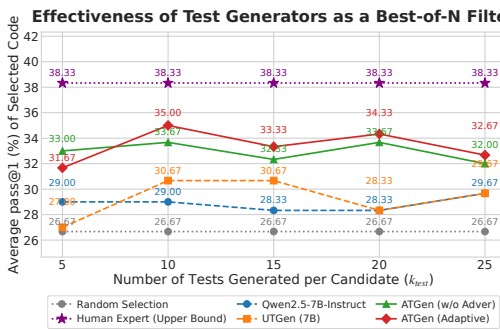

Figure 2: Performance comparison of different test generation models in a Best-of-N code generation setting. ATGEN (Adaptive) achieves the best performance and significantly closes the gap to the theoretical Human Expert upper bound.

Figure 2 shows the average pass@1 of the selected code. The results clearly demonstrate that our adversarially trained model, ATGEN (Adaptive), is the most effective automated filter, consistently outperforming all baselines. At a cost-efficient setting of $k_{test} = 10$, using ATGEN (Adaptive) achieves a final pass@1 of 35.00%, surpassing the UTGen baseline (30.67%) by over 4.3 absolute points and significantly closes the gap to the theoretical Human Expert upper bound (38.33%).

Crucially, the plot also reveals that ATGEN's peak performance is achieved with a small number of generated tests. The performance curves for our models remain largely stable as $k_{test}$ increases beyond 10. We attribute this to the RL objective, which optimizes the generator to find a single, high-impact, bug-revealing test case. Consequently, generating more tests for the same candidate yields no improvements. This demonstrates that ATGEN is not only a powerful verifier but also remarkably compute-efficient, requiring only a few generated tests to reliably select the best candidate from a large pool.

**ATGEN as an RL Reward Source for Code Generation.** Beyond inference, a powerful test generator can provide a reward signal for training code models with RL. Pioneer work like Deepseek-R1 (Guo et al., 2025) has shown that RL, using pass rates on ground-truth test cases as a reward, can dramatically enhance a model's coding ability. However, this approach is limited to problems where a factual test suite already exists. This raises a critical question: can a high-quality generated test suite serve as an effective proxy for RL-based code generation training?

To investigate this, we train a Qwen2.5-3B-Instruct code generator via RL, using reward signals from three different test generators: ATGEN (Adaptive), the baseline UTGen (7B), and a prompted

Figure 3: Final pass@1 performance of a Qwen2.5-3B-Instruct code generator after being trained via RL with rewards provided by different test generators. Using our ATGEN-7B as the reward source yields a substantially stronger final code generator compared to using baseline test generators.

Qwen2.5-7B-Instruct. The resulting code generators are then evaluated on the APPS and Codeforces benchmarks. The results presented in Figure 3 show that the code generator trained with rewards from our ATGEN-7B significantly outperforms the models trained using UTGen or Qwen2.5-7B-Instruct across both benchmarks. This confirms that a high-quality, adversarially trained test generator can effectively proxy human-written test suites in an RL loop, opening a new avenue for improving code models where ground-truth tests are unavailable.

## 5.5 ANALYSIS OF ADVERSARIAL SAMPLING ATTEMPTS

To understand the impact of adversarial pressure, we conduct an ablation study on the maximum number of sampling attempts used to find an adversarial code. Intuitively, the more retires, the more likely an adversarial code could be sampled. We evaluate the performance of our ATGEN-7B model with sampling retries set to 10, 20, and 30.

Table 4: Impact of varying the maximum sampling retries for adversarial code. Increasing retries enhances the adversarial ratio and input attack rate but reduces IO accuracy.

| Max Sampling Retries | Adversarial Code Ratio (%) | IO Accuracy (%) | Input Attack Rate (%) |
|---|---|---|---|
| 10 | 17.1 | 74.09 | 47.96 |
| 20 | 23.0 (↑) | 72.99 (↓) | 49.50 (↑) |
| 30 | 23.7 (↑) | 70.69 (↓) | 49.94 (↑) |

The results, presented in Table 4, reveal two key insights. First, increasing sampling attempts boosts the proportion of adversarial code found, but with diminishing returns. As shown in the "Adversarial Code Ratio" column, an increase from 20 to 30 results in only a marginal improvement on adversarial code ratio. This suggests that as the test generator becomes more adept, the pool of easily discoverable adversarial examples shrinks, making it harder to find new ones even with more attempts. Second, in ATGEN, the adversarial code ratio controls the trade-off between the model's attack capability and its correctness. While the raw bug-finding potential (Input Attack Rate) consistently improves, the model's ability to produce correct outputs (IO Accuracy) steadily declines. These findings highlight that increasing retries is not a monolithic improvement. A moderate number of retries appears to provide a sweet spot, effectively boosting the model's attack capabilities without catastrophically impacting its output accuracy.

## 6 CONCLUSION

This paper introduces ATGEN, a novel framework that trains the test case generator using adversarial reinforcement learning. By creating a dynamic curriculum of challenging bugs, ATGEN learns to produce highly effective test cases that significantly outperform prior methods reliant on static datasets. Our experiments validate ATGEN's superiority and demonstrate its practical utility as a powerful filter for selecting correct code and as a high-quality reward source for training more capable code generation models. This work not only establishes a more effective paradigm for automated debugging but also presents a core adversarial RL framework that is, in principle, generalizable to a broader range of code and reasoning-related challenges.

## ACKNOWLEDGMENTS

We thank all anonymous reviewers and area chairs for their constructive feedback. This work was supported in part by the National Natural Science Foundation of China (62322603, 62502310).

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

## ROLE OF LANGUAGE MODELS FOR THE PAPER

In the process of writing this paper, the language model was used and only used to help us polish the text.

## APPENDIX

## A    ADDITIONAL RESULTS

### A.1    ANALYSIS OF RL TRAINING CURVES FOR CODE GENERATION

To further illustrate the superiority of ATGEN as a reward source for training code generators via reinforcement learning, we present the training curves in Figure 4. These plots provide direct visual evidence supporting the quantitative results presented in the main paper.

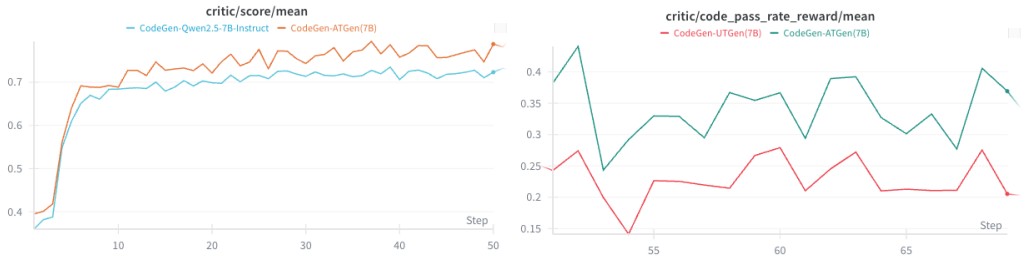

(a) Comparison of mean reward score against a baseline test generator.

(b) Comparison of code pass rate reward against UT-Gen.

Figure 4: Training curves for the code generator when using different test generators as the reward source. (a) The total reward score during the initial 50 steps of training. Using ATGEN leads to a higher and more sustained reward signal. (b) The isolated code pass rate reward after 50 steps. ATGEN provides a significantly more effective learning signal than UTGEN.

Figure 4a compares the mean reward score of a code generator trained using rewards from our ATGEN-7B versus rewards from the baseline Qwen2.5-7B-Instruct test generator. The total reward is an average of three components: a format reward, a tag count reward, and the crucial code pass rate reward. The baseline's training curve quickly plateaus around a score of 0.7. This is because the code generator rapidly masters the simple format and tag-related tasks (achieving a perfect score of 1.0 on both), but the test cases provided by the baseline generator are not effective enough to create a meaningful and optimizable signal for the code pass rate. The learning for this critical component stagnates, capping the average reward. In contrast, the curve for the model trained with ATGEN not only reaches a higher overall score but also shows a continuous upward trend, indicating that our test

generator provides a challenging and consistent learning signal that allows the code generator to keep improving its functional correctness.

Figure 4b provides a more direct comparison by isolating the code pass rate reward in the later stages of training (after 50 steps), comparing ATGEN with the stronger UTGen baseline. The curve for UTGen is consistently low and shows no clear upward trajectory, suggesting that its test cases lack the necessary quality and diversity to drive further improvement in the code generator. Conversely, the reward signal from ATGEN is substantially higher and more dynamic, providing a much more effective curriculum for the code generator to enhance its problem-solving capabilities. These curves unequivocally demonstrate that ATGEN serves as a far more effective reward provider for training advanced code generation models.

## A.2 ABLATION STUDY ON ADVERSARIAL CODE GENERATION STRATEGY

| Method | IO Acc (%) | Attack Rate (%) |
|---|---|---|
| *Sampling-based (Proposed)* | | |
| ATGEN (Unconditional) | 74.97 | 34.57 |
| ATGEN (Adaptive) | 74.09 | **36.22** |
| *Instruction-based (Ablation)* | | |
| ATGEN (Instruct) | 70.47 | 30.73 |
| ATGEN (Instruct, Adaptive) | **76.28** | 32.71 |

Table 5: Comparison of adversarial code generation strategies. All models use the Qwen2.5-7B-Instruct backbone. The sampling-based approach (our proposed method) yields a significantly higher Attack Rate, validating its effectiveness.

To validate our choice of using a sampling-based approach for generating adversarial code, we conduct an ablation study comparing it against a more direct instruction-based method. In the instruction-based setting (ATGEN (Instruct)), we directly prompt the code generator to produce a adversarial code that passes the current test case $T_{\text{gen}}$ while failing the ground-truth test suite $T_{\text{gold}}$. We hypothesize that while this method is computationally cheaper, it risks creating code with "unnatural" or "synthetic" bugs, causing a distributional shift that could negatively impact the test generator's training.

The results are presented in Table 5. The findings confirm our hypothesis. Both instruction-based models show a clear degradation in Attack Rate compared to their sampling-based counterparts. For instance, our proposed ATGEN (Adaptive) achieves a 36.22% Attack Rate, whereas its instruction-based variant, ATGEN (Instruct, Adaptive), only reaches 32.71%. This suggests that training on synthetically generated bugs harms the test generator's ability to identify realistic programming flaws.

Interestingly, the ATGEN (Instruct, Adaptive) model achieves the highest IO Accuracy (76.28%). We attribute this to the nature of reinforcement learning. When faced with noisy and synthetic bugs, optimizing the Attack Rate becomes a more difficult task. Consequently, the model pivots to maximize the reward from the more stable and accessible source: IO Accuracy. Since achieving high IO Accuracy primarily depends on understanding the problem description rather than the buggy code, this objective is unaffected by the noisy training data. The model, therefore, over-optimizes for correct I/O pairing at the expense of its core bug-finding capability. These results strongly justify our use of the more robust sampling-based adversarial generation, as it creates a more realistic and effective training curriculum for the test generator.

## A.3 ANALYSIS OF PRACTICAL UTILITY: PRECISION AND RECALL

While our main evaluation focuses on IO Accuracy and Attack Rate (which corresponds to Recall in bug detection), practical test generation also requires high *Precision* (Liu et al., 2025). A test generator that produces many invalid test cases (False Positives) creates a "crying wolf" scenario, wasting developer time.

Following the definitions in prior work (Liu et al., 2025), we define the metrics as follows:

- **True Positive (TP)**: The test case is valid (correct IO) AND successfully triggers a bug in the faulty code.
- **False Positive (FP)**: The test case raises an alert (claims to find a bug) BUT is itself invalid (incorrect IO). This represents a false alarm.
- **Precision**: $TP/(TP + FP)$. This measures the reliability of the alerts.

- **Recall**: $TP/(TP + FN)$. This is equivalent to our main "Attack Rate" metric.

Table 6 presents the Precision, Recall, and F1 Score on the total test set.

| Method | Precision (%) | Recall (%) | F1 Score (%) |
|---|---|---|---|
| Qwen2.5-7B-Instruct | 19.52 | 12.73 | 15.41 |
| UTGen (7B) | 19.66 | 14.37 | 16.61 |
| ATGEN (w/o Adver) | 65.48 | 34.35 | 45.06 |
| ATGEN (Adaptive) | **70.66** | **36.99** | **47.89** |

Table 6: Precision, Recall, and F1 Score analysis. Precision measures the valid bug-finding signal against false alarms caused by invalid test cases. ATGEN achieves a **3.5x improvement** in Precision compared to the SFT baseline (UTGen).

**Analysis.** The results demonstrate a critical advantage of our RL-based approach. The SFT baseline, UTGen (7B), exhibits a low precision of 19.66%, indicating that a vast majority of its "successful attacks" are actually due to invalid test oracles (e.g., generating incorrect expected outputs that disagree with the buggy code). In contrast, ATGEN (Adaptive) achieves a high precision of **70.66%**.

This **3.5x improvement** in Precision signifies that ATGEN creates a much higher signal-to-noise ratio. The adversarial reinforcement learning process does not merely encourage the model to be aggressive; it forces the model to be *correctly* aggressive. To maximize the reward, the agent learns that it must generate strictly valid tests to successfully penalize the adversary, rather than exploiting invalid inputs. This confirms that ATGEN is not only a powerful bug-finder (High Recall) but also a trustworthy tool for practical software engineering (High Precision).

## A.4 GENERALIZATION

To show the ability of our method's domain generalization and potential overfitting to the training datasets (APPS and Codeforces), we evaluated our trained models on the **TACO** benchmark (Li et al., 2023). TACO represents a completely unseen distribution of coding problems for our model, serving as a rigorous test for zero-shot generalization capability. We used a subset of 1,000 samples from TACO's test set for this evaluation. Table 7 presents the results.

| Method | IO Accuracy (%) | Attack Rate (%) |
|---|---|---|
| Qwen2.5-7B-Instruct | 20.8 | 9.2 |
| UTGen (7B) | 36.1 | 16.5 |
| ATGEN (Adaptive) | **81.1** | **45.9** |

Table 7: Zero-shot generalization results on the unseen TACO benchmark. ATGEN achieves a **45.9% Attack Rate**, nearly tripling the performance of the SFT baseline (UTGen). This confirms that our adversarial RL policy learns a robust, generalizable bug-finding strategy rather than memorizing training data.

**Analysis.** The results unequivocally demonstrate the superior generalization of ATGEN. On this unseen benchmark, the SFT baseline (UTGen) struggles, likely due to its reliance on the static distribution of its training data. In sharp contrast, ATGEN (Adaptive) achieves an **Attack Rate of 45.9%**, which is nearly **triple** that of the baseline.

Furthermore, ATGEN maintains an exceptionally high **IO Accuracy of 81.1%**. This indicates that the adversarial training loop successfully teaches the model a fundamental understanding of code logic and bug detection, rather than merely fitting to the specific patterns of APPS or Codeforces. The model effectively transfers its "attacking" policy to

new domains without any fine-tuning.

## A.5 COST AND EFFICIENCY ADVANTAGE OF ADAPTIVE MODE

Adversarial training involves sampling new buggy codes, which introduces computational overhead. To quantify this cost and justify our design choices, we analyzed the token consumption of our two adversarial modes: *Unconditional* and *Adaptive*.

Table 8 details the total input and output tokens consumed by the adversarial code generator (GPT-4o-mini) over 145 training steps using the Qwen2.5-7B-Instruct backbone.

| Mode | Input Tokens | Output Tokens | IO Acc (%) | Attack Rate (%) |
|------|--------------|---------------|------------|-----------------|
| ATGEN (Unconditional) | 5,865,730 | 29,201,333 | **74.97** | 34.57 |
| ATGEN (Adaptive) | **2,830,685** | **13,461,937** | 74.42 | **36.99** |

Table 8: Cost and performance comparison between Unconditional and Adaptive adversarial modes. ATGEN (Adaptive) consumes **less than half** the tokens of the Unconditional mode while achieving a higher Attack Rate. This confirms that conditionally triggering the adversarial sampling is a highly efficient strategy.

**Analysis.** The comparison reveals a significant efficiency advantage for the Adaptive mode.

- **Token Efficiency**: The Adaptive mode reduces token usage by half and output token usage by **54%** compared to the Unconditional mode.

- **Superior Performance**: Despite the drastically reduced computational cost, the Adaptive mode achieves a higher Attack Rate.

This result supports our hypothesis that focusing computational resources on instances where the current policy is not successful creates a more effective and efficient curriculum than blindly regenerating all training data.

### A.6 BEST-OF-N FILTER FOR GPT-5O-MINI

To demonstrate the generalizability of our test generator and its effectiveness on stronger, next-generation code models, we conducted an additional Best-of-N experiment using **GPT-5o-mini** as the code generator. This model is significantly more capable than the GPT-4o-mini used in our main experiments.

We generated 10 test cases per problem using different test generators to filter $N$ candidate solutions sampled from GPT-5o-mini. The results, reported as Pass@1 on the APPS benchmark across three difficulty levels (Introductory, Interview, Competition), are shown in Table 9.

**Analysis.** The results confirm that AT-GEN remains highly effective even when filtering code from a frontier model like GPT-5o-mini. While baseline methods like UTGen show marginal improvements over random selection or the base model on difficult problems, ATGEN (Adaptive) achieves a distinct advantage.

| Method (Test Generator) | Intro. (Pass@1 %) | Inter. (Pass@1 %) | Comp. (Pass@1 %) |
|-------------------------|-------------------|-------------------|------------------|
| Random Selection | 71 | 62 | 15 |
| Qwen2.5-7B-Instruct | 69 | 73 | 18 |
| UTGen (7B) | 69 | 73 | 19 |
| ATGEN (Adaptive) | **72** | **76** | **23** |

Table 9: Best-of-N filtering performance on candidate solutions generated by **GPT-5o-mini**. ATGEN consistently selects better candidates than baselines. Notably, on the hardest *Competition* level problems, it improves the Pass@1 from 19% (UTGen) to **23%**, demonstrating its ability to detect subtle bugs even in highly capable models.

Most notably, on the **Competition** level problems—which involve complex logic and corner cases—ATGEN improves the Pass@1 to **23%**, surpassing the SFT baseline (19%) by 4 absolute percentage points. This indicates that the adversarial training equips the model to verify and filter sophisticated code that simpler test generators might incorrectly accept.

## B QUALITATIVE CASE STUDY: REASONING VS. MIMICRY

To better understand the qualitative difference between our adversarial approach and standard SFT, we present a representative case study from the Codeforces dataset.

**Problem (Pipe System Connectivity)**: Given a grid, count the number of ways to fill empty cells with pipe segments such that the system is "non-leaking" (all pipe ends connect to another pipe or the border).

**Buggy Code**: The code contains flawed logic in the `is_leaking()` function, specifically failing to correctly handle specific boundary connections for single-cell configurations.

Table 10 compares the test cases generated by the SFT baseline (UTGen) and ATGEN.

| Feature | UTGen (SFT Baseline) | ATGEN (Adversarial RL) |
|---------|----------------------|------------------------|
| Generated Input | `2 2`
`1.`
`..` | `1 1`
`1` |
| Style | **Mimicry**. The model generates a "typical" test case resembling the training distribution (usually larger $N \times M$ grids). | **Targeted Attack**. The model explicitly reasons about the code's logic and identifies a minimal counter-example. |
| Reasoning Process | "Let's break down the problem step by step to understand why the code solution returns 2 for the given input....Thus, the output is 2." | "We can create a simple case where the correct answer is 1, but the buggy code solution returns 2... specific boundary connections..." |
| Result | Failed. The buggy code passes this test (False Negative). | Success. The buggy code fails on this edge case (True Positive). |

Table 10: Qualitative comparison on the "Pipe System Connectivity" problem. While UTGen relies on mimicking the distribution of human-written tests (typically larger grids), ATGEN identifies that the bug lies in a much simpler atomic unit (a $1 \times 1$ grid) and successfully targets it.

**Analysis.** This example highlights the core limitation of static SFT: the model mimics the *distribution* of the training data (human-written tests often use standard $2 \times 2$ or larger grids) rather than analyzing the *code* under test.

In contrast, ATGEN, driven by the adversarial reward, learns to "think" like an attacker. It identifies that the complex logic in `is_leaking()` fails for the simplest boundary condition. By generating a $1 \times 1$ grid input, it successfully exposes the bug that the "typical" test case missed. This demonstrates that our method shifts the paradigm from *pattern matching* to *logical falsification*.

## C    ADDITIONAL RELATED WORK

**Mutation Testing.**    Our adversarial code generation shares conceptual roots with Mutation Testing, a technique that assesses test suite quality by injecting artificial faults (mutants) into programs (Jia & Harman, 2010). However, a key distinction lies in the nature of the bugs. Traditional mutation testing relies on predefined, syntactic operators (e.g., changing $+$ to $-$ or negating conditions) to create "synthetic" mutants. While useful, prior studies have debated whether these synthetic mutants validly represent the complexity of real-world software faults (Just et al., 2014; Andrews et al., 2005). In contrast, ATGEN leverages the generative capabilities of LLMs to sample "natural" bugs. Rather than applying rigid syntactic rules, our adversarial agent generates code with semantic and logical flaws—such as incorrect boundary handling or misinterpreted constraints—that mimic actual mistakes made by developers or AI models Tambon et al. (2025). This ensures that our test generator is trained on a curriculum of realistic challenges rather than artificial artifacts.

**Self-Play RL.**    The adversarial loop in ATGEN can be viewed through the lens of Reinforcement Learning (RL) self-play. Classical self-play frameworks, such as AlphaZero (Silver et al., 2017) or TD-Gammon (Tesauro et al., 1995), typically operate in a *symmetric* zero-sum setting where an agent plays against a copy of itself within an identical state-action space. In contrast, ATGEN establishes an *asymmetric* adversarial game, more akin to the "Setter-Solver" or "Teacher-Student" paradigms explored in identifying intrinsic motivation (Sukhbaatar et al., 2017; Campero et al., 2020). In our framework, the two agents hold distinct roles and objectives: the Code Generator (Setter) aims to construct difficult problem instances (buggy codes), while the Test Generator (Solver) aims to identify their flaws. This asymmetry is essential for our domain, as it allows the "difficulty" of the environment to evolve dynamically based on the semantic complexity of the code, breaking the fixed-difficulty ceiling inherent in static datasets.

# D PROMPTS

In this section, we present the prompts used in training and inference for an LLM to perform various operations.

We present the prompts for generating test case IO pair for test generator in Table 11. And we present the prompts for sampling adversarial code in Table 12. The other prompt we present in Table 13 is the prompt for instructing the code generator to generate adversarial code.

```
Prompt for Generating Test Case IO Pair

system message:
You are a helpful AI Assistant that provides well-reasoned and
    ↪ detailed responses. You first think about the reasoning
    ↪ process as an internal monologue and then provide the
    ↪ user with the answer. Respond in the following format:
    ↪ <think>\n...\n</think>\n<answer>\n...\n</answer>

prompt:
Generate a test case (both input and output) for the given
    ↪ programming problem and buggy code.
The generated input/output pair MUST satisfy the problem
    ↪ description (i.e., the gold code should pass it).
The generated input/output pair MUST cause the provided buggy
    ↪ code to fail (e.g., wrong answer, runtime error).

Format the generated test case within the <answer> tags as a
    ↪ JSON object containing "input" and "output" keys,
    ↪ enclosed in triple backticks '''json '''. For example:
<answer>
'''json
{{
"input": "[your generated test case input]", (Notice: Brackets
    ↪ are not needed in your output)
"output": "[your generated test case output]"
}}
'''
</answer>

Problem:
{example['question']}
Buggy Code:
{example['buggy_code']}
```

Table 11: Prompt for Generating Test Case IO Pair.

```
Prompt for Sampling Adversarial Code

Complete the Python program to solve the problem. Remember to
    ↪ contain the complete program including all the imports
    ↪ and function header in your response.
Generate the code ONLY. No other explanation or words attached!
{question}
```

Table 12: Prompt for Sampling Adversarial Code.

**Prompt for Instruction-based Generation of Adversarial Code**

```
Create Python code that appears to solve the problem but
    ↪ contains subtle flaws. The code MUST:
1. DIRECTLY READ FROM STANDARD INPUT (using input()) and PRINT
    ↪ the result
2. CONTAIN NO EXTRA TEST CODE
3. Pass the specific test case while failing other valid
    ↪ scenarios
4. Include hidden logical errors that aren't obvious at first
    ↪ glance

Example (BAD for illustration):
Problem: "Sum two integers"
Input Format: Two space-separated integers
Bad Solution:
a, b = map(int, input().split())
print(a + b if a < 5 else 0)

Think in <think>\n...\n</think> and answer in
    ↪ <answer>\n...\n</answer>.
The content between <answer> and </answer> should be the code
    ↪ ONLY, without any other words attached.

Problem:
{question}

Test Case (MUST PASS):
{test_case_pair}
```

Table 13: Prompt for Instruction-based Generation of Adversarial Code.

