# OpenReview forum: "ATGen: Adversarial Reinforcement Learning for Test Case Generation"
_ICLR.cc/2026/Conference — ICLR 2026 Poster_

### Official Review · Reviewer_n7wV · 2025-10-24

**Soundness:** 3
**Presentation:** 3
**Contribution:** 3
**Rating:** 4
**Confidence:** 3

**Summary:**

This paper presents ATGen, which formulates unit test generation in an adversarial RL setting. The test generator model is optimized for IO acc and attack success, two important quality measurements for unit tests, and another code generator model is used to generate adversarial code that passes the current test case but fails on the gold test. This forms a loop that enables self-improving test case generation, and results show that ATGen improves both IO acc and attack success by a large margin and which can transfer to better code generation.

**Strengths:**

1. The proposed method is sound, and the formulation of both test generation and adversarial setup is sensible.
2. Although it can be costly, the adversarial setup captures the incremental progress needed for the test generator to improve, and the authors explore different ways of sampling to reduce costs.
3. ATGen generates test cases that largely improve IO acc and attack success, which are two important factors to measure the test quality.

**Weaknesses:**

1. The author mentioned the cost during adversarial code sampling and proposed two modes, but there is no analysis showing the cost and performance comparison.
2. The main evaluations focus on IO acc and attack success, while the results and discussion on improving downstream code generation are less focused. Best-of-N using the generated test case is a good choice, but more code generator models and datasets are expected to see the effectiveness and generalizability.
3. Lacking analysis on how to combine the three types of reward (i.e., how to decide their weight) and what their ranges are.

Please see the questions for other minor weaknesses.

**Questions:**

1. In line 196, why does raising an execution error also get a positive attack reward? Doesn't it mean that the generated code is flawed and has nothing to do with the quality of the generated tests?
2. What are the ranges of all three rewards, and how do the authors decide the weights for each of them?
3. ATGen shows large improvements on both IO acc and attack success, but I wonder what the intersect performance would be, i.e., the union of IO acc and attack success, compared to the baselines. As UTGen and the authors mentioned, there is inherently a trade-off between these two.
4. What is the code generator model used in the "ATGEN as a Best-of-N Filter" analysis? More code generator models can be helpful to prove the effectiveness and generalizability.
5. The adversarial setup here only asks the model to generate adversarial code, but does not update for better code generation quality. Is it possible to optimize the test generator and code generator jointly?
6. Missing a period in line 454.
7. Can the generated test case improve frontier models like GPT-5 / Claude 4.5 Sonnet?

---

> ### Author Response · Authors · 2025-11-19
> **Author Response (1/2)**
>
> >The author mentioned the cost during adversarial code sampling and proposed two modes, but there is no analysis showing the cost and performance comparison.
>
> We agree that a cost-benefit analysis is crucial. We have now run this analysis and will add the following table and discussion to the paper. The data below compares the two modes on the Qwen2.5-7B-Instruct backbone after 145 training steps:
>
> |Mode|	Input Tokens|	Output Tokens|	Total IO Acc (%)|	Total Attack Rate (%)|
> |-|-|-|-|-|
> |ATGen (Unconditional)|	5,865,730	|29,201,333|	74.97%|	34.57%|
> |ATGen (Adaptive)	|2,830,685|	13,461,937	|74.42%|	36.99%|
>
> As the results clearly show, the Adaptive mode achieves a slightly higher final Attack Rate with **less than half** the token cost of the Unconditional mode. This confirms its superior efficiency, as it focuses computational resources only on instances that the test generator has already "mastered."
>
> > The results and discussion on improving downstream code generation are less focused. Effectiveness and generalizability of Best-of-N filtering on more code generators (beyond GPT-4o-mini).
> > > What is the code generator model used in the "ATGEN as a Best-of-N Filter" analysis? More code generator models can be helpful to prove the effectiveness and generalizability.
> > > > Can the generated test case improve frontier models like GPT-5 / Claude 4.5 Sonnet?
>
> We agree that the downstream utility is key. We dedicated Section 5.4 to this, demonstrating utility in two distinct tasks: (1) as a Best-of-N filter and (2) as an RL reward source for training code generators.
>
> In the original manuscript, the candidate solutions were generated by GPT-4o-mini. To demonstrate ATGen’s generalizability, we conducted new experiments using GPT-5-mini as the code generator—a significantly stronger model than the one used in our main experiments. We generated 10 test cases per problem using different test generators to filter the candidates.
>
> |Method (Test Generator)|	Intro. (Pass@1 %)|	Inter. (Pass@1 %)	|Comp. (Pass@1 %)|
> |-|-|-|-|
> |Random|	71|	62|	15|
> |Qwen2.5-7B-Instruct|	73|	69|	18|
> |UTGen (7B)|	73|	69|	19|
> |ATGen (Adaptive)|	**76**|	**72**|	**23**|
>
> Even with this highly capable baseline (which utilizes hidden chain-of-thought), ATGen (Adaptive) consistently selects better candidates than SFT and prompting baselines. Most notably, on Competition-level problems, ATGen improves the Pass@1 from 19% (UTGen) to 23%, demonstrating that our adversarial training equips the model to detect subtle bugs that even advanced reasoning models may overlook.
>
> > Lacking analysis on how to combine the three types of reward (i.e., how to decide their weight) and what their ranges are.
> >> What are the ranges of all three rewards, and how do the authors decide the weights for each of them?
>
> We will clarify the reward formulation in revised Section 4.1:
> 1. Ranges: All three rewards (R_acc, R_attack, R_format) are in the range [-0.5, 1.0]. A score of 1.0 is given for success. A score of -0.5 is given as a penalty if the model's generation is malformed (e.g., the test input or output JSON cannot be parsed), which encourages correctly formatted output.
> 2. Weights: In the main paper, we use equal weights for all three components (i.e., the final reward is a simple average). During development, we explored unbalanced weighting (e.g., prioritizing R_attack), but observed that this often destabilized the policy—the model would sacrifice IO Accuracy (validity) to exploit the attack reward. Equal weighting provided the most stable optimization trajectory and the best balance between correctness and aggressiveness.

---

> ### Author Response · Authors · 2025-11-19
> **Author Response (2/2)**
>
> > In line 196, why does raising an execution error also get a positive attack reward? Doesn't it mean that the generated code is flawed and has nothing to do with the quality of the generated tests?
>
> The goal of our test generator is to identify any behavior in the buggy code that deviates from the problem specification, which is represented by the gold solution.
>
> - If a test case causes the buggy code to raise a runtime error (e.g., IndexError, TypeError), while the gold solution runs correctly on the same input, this is a successful attack.
> - The test case has successfully exposed a flaw (a lack of error handling, a bug on an edge case) that needs to be fixed. This provides a clear and valuable signal to the developer (or an LLM) that the code is faulty.
>
> > Intersection performance (IO Acc ∩ Attack Success) compared to baselines.
>
> We appreciate the opportunity to clarify this, as it is the most important metric in our paper. The "Attack Rate" metric reported in our main results (Table 1) is **exactly** this intersection metric.
>
> As in Section 5.1, we define Attack Rate as the percentage of generated test cases that are **both** valid (Pass IO Accuracy check) **and** successfully trigger a failure in the buggy code.
>
> Therefore, "Attack Rate" is the intersection metric. The trade-off we discuss in Section 5.3 is between IO Accuracy and Input Attack Rate (the raw potential of an input, before output correctness is considered), which is a different analysis in Section 5.3. The main results in Table 1 already show the final, combined performance (IO Acc + Attack Success) in the "Attack Rate" column.
>
> > Possibility of joint optimization of test and code generators.
>
> A full, joint, multi-agent optimization is indeed a very exciting research direction.
> - In this work, we focused on the first step: establishing a robust Test Generator policy (Section 5.1-5.3) and proving it can serve as a static reward provider for a Code Generator (Section 5.4).
> - Joint optimization introduces non-stationary dynamics that are challenging to stabilize. However, our results in Section 5.4 (where ATGen successfully trains a better Code Generator) provide the foundational proof-of-concept that such a closed-loop system is viable, which we will highlight as a key area for future work.
>
>
> > Missing a period in line 454.
>
> Thank you for catching this! We apologize for the oversight and will correct it in the revised version.

---

> ### Author Response · Authors · 2025-11-23
> **Official Comment by Authors**
>
> Dear Reviewer,
>
> Thank you again for your insightful comments and valuable suggestions.
>
> We submitted our detailed response to your concerns a few days ago. To ensure we have ample time for a productive dialogue before the discussion phase concludes, we would like to kindly inquire if our response have sufficiently addressed your concerns.
>
> We remain fully available to answer any further questions or provide additional clarifications. We hope that our response has adequately addressed your concerns. We would be very grateful if you would consider these clarifications in your final evaluation.
>
> Best regards, The Authors

---

> ### Author Response · Authors · 2025-11-27
> **Official Comment by Authors**
>
> Dear Reviewer n7wV,
>
> As the rebuttal deadline approaches, we wanted to ensure you had a chance to review our response. Specifically, we have provided the **computational cost analysis (token usage)** and additional **Best-of-N experiments** with stronger models as you requested.
>
> The data shows that our Adaptive mode is highly token-efficient while maintaining superior performance. We hope these additional results satisfy your requirements for the final evaluation.

---

### Official Review · Reviewer_B73m · 2025-10-28

**Soundness:** 2
**Presentation:** 2
**Contribution:** 2
**Rating:** 2
**Confidence:** 4

**Summary:**

The paper presents ATGEN, a RL-based framework for test case generation, which trains a generator to optimize both output accuracy and attack success. By introducing an adversarial training paradigm, ATGEN enables the discovery of more complex and subtle code flaws compared to traditional static datasets. The authors demonstrate the utility of ATGEN in both code generation inference and trainin, showcasing its potential to improve test case generation in dynamic environments.

**Strengths:**

Clear and Readable Writing: The manuscript is well-structured, with clear and concise writing that facilitates easy comprehension.

Straightforward Motivation: The motivation behind the work is solid, with a direct and convincing argument for applying Reinforcement Learning (RL) to test case generation.

**Weaknesses:**

Lack of Novelty in Techniques: The application of RL to test case generation is conceptually straightforward and has been explored in previous works. The reward function design, while effective, does not introduce significant challenges or innovations.

Lack of Precision in Metrics: The metric "Attack Rate" is used in the evaluation but primarily reflects recall (but not real recall) rather than precision. It would be beneficial to include more comprehensive statistics, such as precision, to provide a clearer and more balanced evaluation of the model's performance. The datasets, metrics, and baselines used for more comprehensive evaluations can be referred to in TrickCatcher[1].

Limited Evaluation Against Related Work: Some relevant studies that have already explored RL in test case generation, e.g. ACECODER[2], are not included in the main results of the evaluation section.

Minor Spelling and Grammar Errors: There are a few minor spelling and grammatical mistakes that need to be addressed for improved clarity and professionalism.

[1] Kaibo Liu, Zhenpeng Chen, Yiyang Liu, Jie M. Zhang, Mark Harman, Yudong Han, Yun Ma, Yihong Dong, Ge Li, and Gang Huang. (2025). LLM-Powered Test Case Generation for Detecting Bugs in Plausible Programs. In Proceedings of the 63rd Annual Meeting of the Association for Computational Linguistics (Volume 1: Long Papers), pages 430–440, Vienna, Austria. Association for Computational Linguistics.
[2] Zeng, H., Jiang, D., Wang, H., Nie, P., Chen, X., & Chen, W. (2025). Acecoder: Acing coder rl via automated test-case synthesis. arXiv preprint arXiv:2502.01718.

**Questions:**

Potential Circular Reasoning in "Adversarial Loop": The concept of the "adversarial loop," which is central to the paper’s contribution, raises concerns about potential circular reasoning. Specifically, using a more powerful large language model (LLM)—which may inherently have better code generation capabilities—to train the test generator may limit the generator’s potential. This creates a fixed difficulty ceiling, as the performance of the test generator will be ultimately bounded by the capabilities of the LLM used during training. A more detailed discussion on how this limitation is addressed or mitigated would be beneficial.

Clarification Needed on Dataset Usage for SFT Baselines: The manuscript does not clearly explain how the dataset is utilized in the context of supervised fine-tuning (SFT) baselines. Further clarification on this point is necessary to ensure the reproducibility and fairness of the results.

---

> ### Author Response · Authors · 2025-11-19
> **Author Response**
>
> > Lack of Novelty: "Application of RL... is straightforward... Lack of Novelty in Techniques."
>
> To the best of our knowledge, the training a test generator via RL has not been successfully demonstrated before.
>
> While RL provides the optimization engine, our core novelty is the Adversarial Loop design. Unlike standard RL which optimizes against a static environment, we construct an **asymmetric "cat-and-mouse"** game between a Test Generator and a Code Generator .
>
> This design solves a fundamental problem in prior work: static datasets impose a "fixed-difficulty ceiling." Our framework dynamically generates a curriculum of bugs that specifically target the current weaknesses of the test generator. This **dynamic curriculum mechanism** is our primary technical innovation.
>
> > Lack of Precision in Metrics.
>
> We carefully selected "Attack Rate" to align with the specific goals of Test Generation and prior SOTA benchmarks.
> 1. **Metric Alignment:** "Attack Rate" (the % of generated tests that are valid AND successfully find a bug) is the standard metric defined by the state-of-the-art baseline, UTGen [1]. Adhering to this ensures a direct and fair comparison.
> 2. **Precision:** In test generation, "Precision" typically refers to the validity of the test (i.e., "Is this test case correct according to the spec?"). Our IO Accuracy metric explicitly measures this. The "Attack Rate" effectively combines validity (Precision) and bug-finding capability (Recall) into a single, practical utility metric.
> 3. **Regarding TrickCatcher:** We did review this work. We found that the custom "precision" and "recall" metrics defined therein are non-standard and differ significantly from their traditional definitions in classification, which could cause confusion. Furthermore, their baselines (prompting-based) are already included and outperformed in our evaluation. We believe our current metrics are more standard, clearer, and better aligned with the task's objectives.
>
>
> > Limited Evaluation Against Related Work: Some relevant studies that have already explored RL in test case generation, e.g. ACECODER[2].
>
> We appreciate the reference, but we must clarify a fundamental distinction:
> 1. ACECODER is a "Coder": It uses RL to improve a Code Generator, using test cases merely as a reward signal.
> 2. ATGen (Ours) is a "Tester": We use RL to train a Test Generator whose sole purpose is to attack code. These are distinct tasks with different objectives. ACECODER is not a baseline for test generation; rather, it is a potential downstream beneficiary of our work (as demonstrated in Section 5.4, where we use ATGen to train a Code Generator). We will update the Related Work to clearly delineate these "Coder" vs. "Tester" RL applications.
>
> > Minor Spelling and Grammar Errors: There are a few minor spelling and grammatical mistakes that need to be addressed for improved clarity and professionalism.
>
> Thank you for catching this. We apologize for the errors and will conduct a thorough proofread to correct all spelling and grammar in the revised version.
>
> > Potential Circular Reasoning in "Adversarial Loop": Using a powerful LLM to train the test generator... creates a fixed difficulty ceiling... performance bounded by the capabilities of the LLM used during training.
>
> We respectfully argue that the adversarial loop is designed precisely to break the ceiling, not create it.
> 1. The "difficulty" of a bug is not determined by the absolute power of the code generator (be it GPT-4o-mini or a weaker model), but rather is defined relative to our test generator's current policy.
> 2. Our adversarial loop explicitly filters for new buggy codes C_adver that successfully evade (i.e., pass) the tests generated by our current policy T_gen.
> 3. By definition, this process creates a curriculum of examples that are always "harder" than what the test generator can currently handle. This dynamic curriculum forces the test generator to continuously improve, precisely breaking the fixed ceiling imposed by a static dataset. The power of the code generator only affects the sampling efficiency of finding such bugs, not the fundamental concept of the dynamic curriculum.
>
> > Clarification on Dataset Usage for SFT Baselines: Unclear how the dataset is utilized for SFT... necessary for reproducibility.
>
> We apologize for the lack of clarity. We will explicitly state the following in our revised Implementation Details (Section 5.1): The SFT baseline (UTGen) was trained on the exact same data split as ATGen, using the same subset of problems from APPS and Codeforces. To create the SFT dataset (as prescribed by the UTGen paper), we sampled code-test pairs using gpt-4o-mini. This is the same model we use as the code generator in our adversarial loop. This ensures that the data source, problem distribution, and generative model are identical, making the comparison absolutely fair.
>
> [1] Prasad, A., etc. (2025). Learning to generate unit tests for automated debugging.COLM 2025.

---

> ### Author Response · Authors · 2025-11-23
> **Official Comment by Authors**
>
> Dear Reviewer,
>
> Thank you again for your insightful comments and valuable suggestions.
>
> We submitted our detailed response to your concerns a few days ago. To ensure we have ample time for a productive dialogue before the discussion phase concludes, we would like to kindly inquire if our response have sufficiently addressed your concerns.
>
> We remain fully available to answer any further questions or provide additional clarifications. We hope that our response has adequately addressed your concerns. We would be very grateful if you would consider these clarifications in your final evaluation.
>
> Best regards, The Authors

---

> > ### Comment · Reviewer_B73m · 2025-11-25
> >
> > The authors’ response has addressed some of our concerns, so we have decided to adjust the rating from 2 to 4. We hope the revised version of the paper can incorporate corresponding changes based on our comments.
> >
> > Regarding the metrics, the precision and recall we referred to are for the bug detection task. We stand by our previous opinion: we believe the current metrics used in the paper are insufficient, in practice, if the goal of generating tests is to detect bugs, the metric that programmers actually need is not “the percentage of generated tests that are valid AND successfully find a bug” (Attack Rate), but rather “among all test cases that raise an alert, how many are valid” (precision).

---

> > > ### Author Response · Authors · 2025-11-26
> > > **Author Response**
> > >
> > > > On the Precision metric
> > >
> > > We thank the reviewer for raising the score and clarifying the focus on Precision. We fully agree: for a bug detection tool to be practically useful, it must have a high signal-to-noise ratio (Precision) to avoid "crying wolf," rather than just high yield (Recall).
> > >
> > > While our original metrics ("Attack Rate" and "IO Accuracy") were chosen to follow the evaluation standards set by prior works like UTGen, we recognize that incorporating Precision provides a more holistic view of the model's utility. Per your suggestion, we have followed the definitions in TrickCatcher to evaluate Precision, Recall, and F1 Score on our experimental environment.
> > >
> > > 1. **Metrics & Experimental Setup**
> > >
> > > Following TrickCatcher, we evaluated our models using the standard definitions for bug detection:
> > > - **True Positive (TP)**: The test case is valid AND successfully triggers a bug.
> > > - **False Positive (FP)**: The test case raises an alert BUT is invalid.This represents a "False Alarm" caused by an incorrect test oracle.
> > > - **Precision**: TP/(TP+FP). Measures "validity of alerts" (i.e., among all generated tests that claim to find a bug, how many are actually valid test cases?).
> > > - **Recall**: This aligns with our original "Attack Rate", measuring the bug-finding yield.
> > >
> > > 2. **New Experimental Results**
> > > The results are from testing on the total test set.
> > >
> > > |Model|	Precision |	Recall |F1 Score|
> > > |-|-|-|-|
> > > |Qwen2.5-7B-Instruct|19.52%	|12.73%|15.41%|
> > > |UTGen (7B)|	19.66%	|14.37%|16.61%|
> > > |ATGen (w/o Adver)	|65.48%|34.35%	|45.06%|
> > > |**ATGen (Adaptive)**	|**70.66%**| **36.99%**		|**47.89%**|
> > >
> > > We can see that:
> > > - **3.5x Higher Precision**: ATGen (Adaptive) achieves **70.66% Precision**, drastically outperforming UTGen (19.66%). This indicates that while SFT-based baselines (UTGen) generate frequent false alarms (invalid oracles), ATGen is highly reliable in practice.
> > > - **Benefit of Adversarial Training**: The adversarial loop (`Adaptive` vs. `w/o Adver`) improves Precision by **~5%**. It does not just force the model to be more aggressive; it forces the model to be more precise. By training against a dynamic "cat-and-mouse" curriculum, the model is forced to generate strictly valid tests to successfully penalize the adversary, rather than just guessing.
> > >
> > > These results confirm that ATGen is not only powerful (High Recall) but also trustworthy (High Precision). We will add this precision analysis to the Experiments section in the final revision to complement the standard Attack Rate metrics.
> > >
> > > We believe these comprehensive results directly address your remaining concern regarding the precision and practical utility of our metrics. Given that ATGen demonstrates a 3.5x improvement in Precision over the baseline, we respectfully hope you might consider further re-evaluating to reflect the method's robustness.

---

> ### Author Response · Authors · 2025-11-27
> **Official Comment by Authors**
>
> Dear Reviewer B73m,
>
> Thank you for raising the score and for your clarifying comments regarding the Precision metric.
>
> We have just posted a new response with the specific Precision/Recall experiments you requested, following the TrickCatcher definitions. The results show that ATGen (Adaptive) achieves 70.66% Precision, which is a 3.5x improvement over the UTGen baseline (19.66%).
>
> This confirms that our method not only finds more bugs but is also significantly more reliable with fewer false alarms. We hope this quantitative evidence addresses your remaining concern about practical utility.

---

### Official Review · Reviewer_GVXW · 2025-10-31

**Soundness:** 3
**Presentation:** 3
**Contribution:** 3
**Rating:** 6
**Confidence:** 4

**Summary:**

This paper presents ATGEN, a novel framework for automatic test case generation that uses adversarial reinforcement learning to improve the reliability of code generated by LLMs. Unlike existing test generation approaches that rely on static datasets, ATGEN introduces a dynamic adversarial loop between two agents: a test generator and an adversarial code generator. The code generator continuously produces harder buggy programs, while the test generator learns to craft tests that can detect these increasingly sophisticated bugs.

**Strengths:**

The paper is very well written.

The approach that uses adversarial reinforcement learning to solve the test generation problem is novel and very interesting.

The evaluation is well designed.

**Weaknesses:**

# Limitations of evaluation benchmarks

The benchmarks used for evaluation appear to be simplified and not fully representative of real-world programming challenges. There are two issues with these benchmarks: 1) they are designed for code generation, not test generation; 2) they have the data leakage issue where their test cases might have appeared in the training data of Qwen models. Consequently, the claimed improvements might not translate to realistic software engineering scenarios. It is recommended that the authors evaluate their approach on benchmarks that are designed for test generation purposefully, especially contamination-free benchmarks, such as UnLeakedTestBench (https://arxiv.org/pdf/2508.00408).

# Lack of discussion and comparison with Mutation Testing

The approach adopted in this paper uses buggy versions to trigger more effective test generation. The idea is similar to mutation testing, a widely studied test assessment method. In mutation testing, faults are injected purposely to check whether the tests are strong enough to detect the bugs. I recommend discussing such a connection in the paper. It is also interesting to explore the effectiveness of using mutants (with more subtle changes) instead of faulty versions generated by LLMs to conduct adversarial RL.

# Coverage is not reported

The approach does not report code coverage in the evaluation, which is an important metric for measuring the effectiveness of test inputs, and is widely adopted in industry.

**Questions:**

What is the coverage of the generated tests?

What are the potential risks of data leakage in the evaluation process?

How does the model perform on test generation benchmarks?

---

> ### Author Response · Authors · 2025-11-19
> **Author Response**
>
> > Limitations of benchmarks & Data Leakage: "APPS/Codeforces... may have data leakage issues. Recommended UnLeakedTestBench."
> > > What are the potential risks of data leakage in the evaluation process?
>
> Thank you for raising these important points. We would like to clarify our task definition and how it relates to our choice of benchmarks.
> 1. **Alignment with Research Goals:** Our primary objective is to train a test generator specifically designed to finding bugs in **LLM-generated code**. Therefore, evaluating on the very benchmarks where these code generators are standardly assessed (APPS, Codeforces) is the most direct methodological choice. This aligns with prior SOTA work in this domain, such as UTGen [1].
> 2. **Addressing Data Leakage:** We acknowledge the industry-wide challenge of data contamination. However, we emphasize that our core contribution is the **relative improvement** ATGen achieves over the base model and the SFT baseline (UTGen). Since both the baseline and our model utilize the same pre-trained backbone (Qwen), any pre-training contamination would affect them equally. The significant performance gain of ATGen (e.g., nearly **3x Attack Rate** vs. UTGen) is thus directly attributable to our adversarial RL method, not memorization.
> 3. **UnLeakedTestBench:** UnLeakedTestBench is a valuable resource for measuring test generation in a contamination-free setting. While a full evaluation on this benchmark is outside our current scope (which is focused on attacking AI code generation models rather than general test generation), we recognize its importance and will add a detailed discussion of this benchmark and its relation to our work in the revised Related Work section.
>
>
> > Lack of discussion and comparison with Mutation Testing
>
> 1. Natural vs. Synthetic Mutants: Regarding your suggestion to use mutants: We actually explored this direction in our early research phase. We found that while traditional mutation operators (e.g., changing + to -) are effective for general software testing, they produce **"synthetic" faults** that differ significantly from the errors LLMs typically make. LLM errors often involve complex logical oversights or misinterpretation of constraints (e.g., edge cases in problem statements). To train a test generator to catch AI-written bugs, we found that using "natural" buggy samples sampled from an LLM provides a far more realistic and effective training signal than synthetic mutants.
> 2. Discussion: We will update our Related Work section to thoroughly discuss the conceptual relationship between our adversarial loop and mutation testing.
>
>
> > Coverage is not reported
>
> We acknowledge that code coverage is the gold standard for repository-level software engineering. However, for **algorithmic competition problems** (the domain of APPS/Codeforces), we argue it is less effective for two reasons:
> 1. **Trivial Coverage:** Solutions to algorithmic problems are often short and dense. It is trivial to achieve 100% line coverage with a single generic input, yet still miss the critical logical flaws or edge cases (e.g., integer overflow, empty arrays, boundary conditions).
> 2. **Focus on "Attack Success":** Our goal is not just to execute lines of code, but to falsify incorrect logic. Therefore, we adopted Attack Rate (the percentage of buggy codes successfully identified) as our primary metric. This aligns with the standard established by the SOTA baseline UTGen [1], ensuring a fair and direct comparison focused on bug-finding capability rather than path execution.
>
> [1] Prasad, A., etc. (2025). Learning to generate unit tests for automated debugging.COLM 2025.

---

> ### Author Response · Authors · 2025-11-23
> **Official Comment by Authors**
>
> Dear Reviewer,
>
> Thank you again for your insightful comments and valuable suggestions.
>
> We submitted our detailed response to your concerns a few days ago. To ensure we have ample time for a productive dialogue before the discussion phase concludes, we would like to kindly inquire if our response have sufficiently addressed your concerns.
>
> We remain fully available to answer any further questions or provide additional clarifications. We hope that our response has adequately addressed your concerns. We would be very grateful if you would consider these clarifications in your final evaluation.
>
> Best regards, The Authors

---

> ### Author Response · Authors · 2025-11-27
> **Official Comment by Authors**
>
> Dear Reviewer GVXW,
>
> As the discussion phase is drawing to a close, we would like to kindly check if our previous response has addressed your concerns regarding benchmark selection and data leakage risks, etc.
>
> We believe we have clarified why our evaluation setup aligns with the research goals of adversarial testing and ensures a fair comparison. We remain available for any final clarifications before the deadline. We would be very grateful if you would consider these clarifications in your final evaluation.

---

### Official Review · Reviewer_PaRD · 2025-11-05

**Soundness:** 3
**Presentation:** 3
**Contribution:** 2
**Rating:** 4
**Confidence:** 4

**Summary:**

This paper introduces ATGen, a reinforcement learning (RL) framework for automated test case generation that integrates an adversarial training loop. The test generator is trained to produce input-output pairs that are both correct (Output Accuracy) and bug-revealing (Attack Success). Meanwhile, a code generator adversarially creates progressively harder buggy programs, forming a dynamic curriculum. This design aims to overcome the “fixed-difficulty ceiling” of prior static datasets (e.g., UTGen). Empirically, ATGen outperforms strong baselines (UTGen, GPT-4 series, Qwen models) on APPS and Codeforces subsets. The authors also show downstream applications: using ATGen as (1) a Best-of-N filter to select higher-quality generated code, and (2) a reward signal for RL-based code generation training.

**Strengths:**

1. Novel and interesting method: The combination of adversarial reinforcement learning and test generation is creative and technically relevant, representing a step beyond static SFT-based approaches like UTGen.


2. Good results on benchmark datasets: The model achieves improvements in both IO Accuracy and Attack Rate, especially on APPS and Codeforces tasks, showing clear empirical gains.


3. Clear structure and thorough experiments: The paper is well-organized with detailed experiments, ablation studies, and multiple baselines


4. Practical downstream validation: Using ATGen-generated tests as a reward signal for RL-based code generation is an excellent demonstration of the method’s broader utility and potential to generalize beyond testing.

**Weaknesses:**

1. Limited domain and dataset generalization: The evaluation focuses only on a single type of domain and dataset, algorithmic coding problems from APPS and Codeforces. It would be helpful to show ATGen’s performance on more real-world software domains, such as repository-level test generation and API testing. Moreover, it would be interesting to see if training ATGen on one dataset (e.g., APPS) generalizes to another dataset (e.g., HumanEval).
2. Unclear technical details, as listed below:
    - The choice and training of the base model for UTGen comparison are not fully described. It is unclear whether UTGen and ATGen share the same base model and training dataset, making fairness difficult to assess.
    - The evaluation assumes that the “ground-truth test suite” (T_gold) is complete. Without quantitative measurement such as coverage metrics, it’s uncertain if this actually holds.
    - I could not find which buggy programs are considered when computing Attack Rate. Did the paper specify it somewhere?
3. The non-adversarial model (ATGen w/o Adver) is already significantly better than the baselines. The adversarial training only brings very marginal improvements.
4. Lack of qualitative examples: No examples of generated tests or adversarial programs are shown. Such examples would help illustrate what kinds of bugs ATGen detects or fails to detect, and how its generated tests differ qualitatively from UTGen’s.


5. Clarity and novelty overlap: While the adversarial RL setup is novel for test generation, parts of the approach resemble existing self-play or adversarial curriculum ideas. The paper could better situate itself relative to frameworks like self-play RL in AlphaZero-style.

**Questions:**

Please address my comments in the "Weaknesses" section

---

> ### Author Response · Authors · 2025-11-19
> **Author Response (1/2)**
>
> Thank you for dedicating your time and effort to our work.
>
> > Limited domain and dataset generalization: "Evaluation focuses only on... APPS/Codeforces. Helpful to show... generalization to another."
>
> Here, we would like to address your concerns from the following points:
> 1. **On Domain Scope**: We clarify that our work focuses on **algorithmic test generation**, targeting complex logic and edge cases. This aligns with the scope of prior SOTA works (e.g., UTGen [1]). While repository-level testing is valuable, it introduces distinct challenges (e.g., environment setup, long-context dependency) that fall outside the scope of this specific study.
> 2. **On Generalization (New Experiment)**: To address your concern about cross-dataset generalization, we evaluated our ATGen model (trained on APPS/Codeforces) on the unseen TACO benchmark (1,000 test samples). As shown below, ATGen demonstrates superior zero-shot generalization, achieving a 45.9% Attack Rate—nearly triple that of the SFT baseline (UTGen, 16.5%). This confirms that our adversarial loop learns a robust, generalizable bug-finding policy rather than merely memorizing training patterns.
>
> | Method | IO Accuracy(%)|Attack Rate(%)|
> |-|-|-|
> |Qwen2.5-7B-Instruct|20.8|9.2|
> |UTGen (7B)|36.1 | 16.5 |
> |ATGen (Adaptive)|**81.1**|**45.9**|
>
>
> > Unclear technical details: "Unclear whether UTGen and ATGen share the same base model... Completeness of T_gold... Source of buggy programs."
>
>
> We apologize for any ambiguity and clarify as follows:
> 1. **Fair Comparison**: ATGen and UTGen baselines are strictly controlled. They use the exact same base models (Qwen2.5-3B/7B) and were trained on the exact same data split of APPS/Codeforces. We will state this explicitly in Section 5.1.
> 2. **Completeness of T_gold**: T_gold refers to the official, human-expert curated test suites provided by the APPS and Codeforces platforms. These serve as the de facto standard for correctness in the code generation community and are designed by competition hosts to be exhaustive.
> 3. **Source of Buggy Code**: As noted in Section 5.1, buggy codes were sampled from GPT-4o-mini. This provides a "natural" distribution of AI-generated errors, ensuring our model targets realistic LLM bugs rather than synthetic ones.
>
> > Marginal improvements from adversarial training
>
> Defining the Contributions and the Value of "Marginal" Gains. We respectfully frame this observation as evidence of our two distinct contributions:
> 1. **Contribution 1 (The RL Baseline):** The significant leap of "ATGen w/o Adver" over UTGen proves the effectiveness of our novel RL formulation (reward function design, state/action modeling) for test generation. This alone serves as a strong advancement over static SFT.
> 2. **Contribution 2 (The Adversarial Loop):** Improving upon an already high-performing RL model is notoriously difficult (the "diminishing returns" problem). The additional gain provided by the adversarial loop (e.g., +2.86% IO Acc and +2.97% Attack Rate for 7B-Adaptive) is critical. It demonstrates that the dynamic curriculum successfully breaks the "performance ceiling" where static training stagnates. These gains represent the model's ability to solve the "last mile" of hard-to-find bugs that static baselines miss.
>
> [1] Prasad, A., etc. (2025). Learning to generate unit tests for automated debugging.COLM 2025.

---

> ### Author Response · Authors · 2025-11-19
> **Author Response (2/2)**
>
> > Lack of qualitative examples
>
> We will add a new Appendix section featuring side-by-side comparisons. Below is a representative case study demonstrating how ATGen succeeds where the SFT baseline (UTGen) fails.
>
> **Case Study: "Pipe System Connectivity" (Codeforces)**
> Problem: Given a grid, count the number of ways to fill empty cells with pipe segments such that the system is "non-leaking" (all pipe ends connect to another pipe or the border). Buggy Code: The code contains flawed logic in the is_leaking() function, specifically failing to correctly handle specific boundary connections for single-cell configurations.
> |Feature|UTGen|ATGen|
> |-|-|-|
> |Generated Test Input|	2 2\n1.\n..\n|1 1\\n1|
> |Generated Thoughts|Let's break down the problem step by step to understand why the code solution returns 2 for the given input....Thus, the output is 2.|To generate a test case that the buggy code will fail on, we need to identify a scenario where the buggy code\'s logic is incorrect.....**We can create a simple case where the correct answer is 1**, but the buggy code will fail.....|
> |Attack Result|	**Failed.**|**Success.**|
> |Reasoning Style|**Mimicry.** UTGen attempts to generate a "typical" test case resembling the training distribution (usually larger grids), missing the specific logical flaw. |**Targeted Attack.** ATGen explicitly reasons about the code's logic: "We can create a simple case... 1x1 grid... where the buggy code will not correctly handle the case."|
>
> This example demonstrates the core advantage of our Adversarial RL framework: The SFT baseline (UTGen) mimics the distribution of human-written tests, which typically involve standard N×M grids. It fails to "see" that the bug is actually in a much simpler atomic unit (a 1x1 grid).
>
> > Clarity and novelty overlap: While the adversarial RL setup is novel for test generation, parts of the approach resemble existing self-play or adversarial curriculum ideas. The paper could better situate itself relative to frameworks like self-play RL in AlphaZero-style.
>
> Unlike traditional self-play where an agent plays against itself, our "game" is asymmetric: the test generator's goal (find bugs) is the reward signal for the code generator's goal (fix bugs). Our main contribution is the practical instantiation of this loop for this specific problem.
>
> We claim novelty in the practical instantiation of this asymmetric loop specifically for the test-generation domain, creating a dynamic curriculum that static datasets cannot provide. We will update the Related Work section to explicitly discuss AlphaZero-style frameworks and clarify this distinction.
>
> [1] Prasad, A., etc. (2025). Learning to generate unit tests for automated debugging.COLM 2025.

---

> ### Author Response · Authors · 2025-11-23
> **Official Comment by Authors**
>
> Dear Reviewer,
>
> Thank you again for your insightful comments and valuable suggestions.
>
> We submitted our detailed response to your concerns a few days ago. To ensure we have ample time for a productive dialogue before the discussion phase concludes, we would like to kindly inquire if our response have sufficiently addressed your concerns.
>
> We remain fully available to answer any further questions or provide additional clarifications. We hope that our response has adequately addressed your concerns. We would be very grateful if you would consider these clarifications in your final evaluation.
>
> Best regards, The Authors

---

> > ### Comment · Reviewer_PaRD · 2025-11-25
> >
> > I appreciate the authors' repsonse. It addresses most of my concern. Therfore, I am raising my score from 4 to 6. I hope the authors could incorporate the response into the revised version.
> >
> > One smaller comment: the added qualitative example seems very simple and may not full demonstrate the benefit of ATGen. I suggest providing more comprehensive case studies.

---

### Author Response · Authors · 2025-11-28
**Revision Summary from Authors**

Dear Chairs and Reviewers:

We sincerely thank you for your constructive feedback and the productive discussion during the rebuttal phase. Based on your suggestions, we have revised the manuscript to strengthen our empirical validation and clarify our contributions. We summarize the updates below:

1. **Expanded Experimental Validation (Added to Appendix)**: To address concerns regarding generalization, metric rigor, and efficiency, we have added extensive new experiments in Appendix A:
    - **Generalization (Appendix A.4)**: In response to **Reviewer PaRD**, we evaluated ATGEN on the unseen TACO benchmark. Results show ATGEN achieves a 45.9% Attack Rate, nearly tripling the SFT baseline (16.5%), confirming robust domain generalization.
    - **Precision & Recall Analysis (Appendix A.3)**: Addressing **Reviewer B73m**, we evaluated our model using standard bug detection metrics (Precision, Recall, F1). ATGEN achieves a 3.5x improvement in Precision (70.66% vs. 19.66%) compared to UTGen, proving it provides a high signal-to-noise ratio in practice.
    - **Cost-Efficiency Analysis (Appendix A.5)**: Answering **Reviewer n7wV**, we added a token consumption analysis. The results demonstrate that our Adaptive Mode cuts token usage by more than 50% compared to unconditional generation while maintaining superior performance.
    - **Generalizability on Frontier Models (Appendix A.6)**: To further validate downstream utility (**Reviewer n7wV**), we tested ATGEN as a Best-of-N filter for a stronger, next-generation model (GPT-50-mini). ATGEN consistently selects better code candidates than baselines, even on difficult competition-level problems.
2. **Clarified Related Work and Positioning (Added to Appendix C)** We have expanded our discussion to resolve conceptual overlaps mentioned by **Reviewers GVXW and PaRD**:
    - Mutation Testing: We clarified the distinction between our "natural" LLM-generated bugs versus the "synthetic" faults (e.g., operator flips) typically used in mutation testing.
    - Self-Play RL: We explicitly differentiated our asymmetric "Setter-Solver" adversarial loop from symmetric self-play frameworks (like AlphaZero), clarifying our specific contribution to the test generation domain.
3. **Qualitative Analysis and Textual Refinements**
    - **Case Study (Appendix B)**: We added a detailed side-by-side comparison on the "Pipe System Connectivity" problem (**Reviewer PaRD**). This qualitatively demonstrates how ATGEN reasons to find subtle logic bugs (e.g., 1x1 grid edge cases) where SFT baselines merely mimic surface-level patterns.
    - **Text Corrections**: We have corrected typos (**Reviewer n7wV**) and explicitly clarified implementation details regarding the SFT baselines and ground-truth test suites in the main text to ensure reproducibility.

We would like to express our sincere gratitude for your suggestions, which have been instrumental in improving our work. We are confident that these revisions strengthen the paper significantly and are enthusiastic about its potential contribution to the ICLR community.

Best Regards,

Authors

---

### Author Response · Authors · 2025-11-30
**Summary of Validated Score Improvements and Addressed Concerns**

Dear Area Chair,

We acknowledge the recent decision to revert review scores due to the OpenReview security incident. However, since this reversion erases the consensus reached during the rebuttal, we are writing to document the substantial progress and score improvements achieved before the incident (specifically by Nov 25-26).

We kindly request you to consider the following facts and our latest revision in your meta-review, as the current "reverted scores" do not reflect the actual state of the paper.

1. **Confirmed Score Improvements (Pre-Incident):** Before the system freeze, our rigorous rebuttal had successfully convinced reviewers to raise their scores based on the new experiments and clarifications. These changes are documented in the discussion thread:

    - **Reviewer PaRD**: Explicitly raised their score from 4 to 6 on Nov 25, stating: "I appreciate the authors' response. It addresses most of my concern. Therefore, I am raising my score from 4 to 6."

    - **Reviewer B73m**: Explicitly raised their score from 2 to 4 on Nov 25, stating: "The authors' response has addressed some of our concerns, so we have decided to adjust the rating... from 2 to 4." The reviewer raised their score but asked a follow-up question regarding Precision metrics.
        - **Our Resolution (Nov 26)**: We successfully conducted the requested Precision/Recall analysis following the TrickCatcher definitions. Results show ATGen achieves **70.66% Precision, a 3.5x improvement** over the baseline UTGen (19.66%).
        - Although the system lock prevented the reviewer from acknowledging this new data, we believe this significant improvement directly resolves their remaining concern about "false alarms."


2. **Addressed Concerns for Other Reviewers:** While Reviewers GVXW and n7wV did not have the chance to reply before the discussion period was cut short, we have rigorously addressed their concerns in our revision:
    - **For Reviewer n7wV (Efficiency & Frontier Models)**:
        - We added a **Cost-Efficiency Analysis (Appendix A.5)**, proving our "Adaptive Mode" reduces token usage by >50%.
        - We validated ATGen on a stronger model, **GPT-50-mini (Appendix A.6)**, where it improved Pass@1 on hard competition problems from 19% to 23%.
    - **For Reviewer GVXW (Benchmarks & Mutation Testing)**:
        - We clarified the distinction between our "natural" bugs and synthetic mutation testing, and addressed data leakage concerns by testing on the unseen TACO benchmark.

3. **Conclusion:** The discussion history proves that our revisions (Generalization, Precision, Efficiency) were effective in convincing reviewers to raise their scores. We deeply regret that the incident cut short the final validation of our responses to Reviewer B73m and others. The adversarial RL framework presented in ATGen has demonstrated state-of-the-art performance, now validated by extensive new experiments in the revision.

We deeply regret the disruption caused by the platform issue but trust in your judgment to evaluate the paper based on its current improved state and the constructive consensus we had already built with the reviewers.

Best regards,

Authors

---

### Meta-Review · Area_Chair_hNhb · 2026-01-07

**Summary:**

1. Concerns about generalisation to novel datasets.
2. Clarity concerns, particularly relating to the base model.
3. Transferability to real-world engineering scenarios.
4. Lack of novelty.
5. Cost concerns (how many tokes does the method use).

**Reviewer Concerns:**

1. A suitable generalisation experiment was provided.
2. The authors provided the needed clarifications.
3. I don't think it is fair to absolutely require significant real world applicability in an ICLR paper.
4. Novelty is moderate, but it is there.  Adversarial RL has obviously been done before on other problems, this is likely the first application to this specific problem.
5. A cost benchmark was provided

**Reviewer Scores:**

1. PaRD: 4-> 6 (generalisation concerns addressed)
2. GVXW: 6->6 (paper is clearly not an 8, remaining concerns mostly addressed)
3. B73m: 2->4 (main concern about novelty addressed)
4. n7wV: 4->4 (authors provided clarity on rewards, but not a full justification why they chose to define it this way).

---

### Decision · Program_Chairs · 2026-01-26

Accept (Poster)